# Genomic and Pathological Characterization of Acute Hepatopancreatic Necrosis Disease (AHPND)-Associated Natural Mutant *Vibrio parahaemolyticus* Isolated from *Penaeus vannamei* Cultured in Korea

**DOI:** 10.3390/ani14192788

**Published:** 2024-09-26

**Authors:** Ye Bin Kim, Seon Young Park, Hye Jin Jeon, Bumkeun Kim, Mun-Gyeong Kwon, Su-Mi Kim, Jee Eun Han, Ji Hyung Kim

**Affiliations:** 1Department of Food Science and Biotechnology, College of Bionano Technology, Gachon University, Seongnam 13120, Republic of Korea; zxc5620@gachon.ac.kr; 2Laboratory of Aquatic Biomedicine, College of Veterinary Medicine and Research Institute for Veterinary Science, Seoul National University, Seoul 08826, Republic of Korea; lovesun139@snu.ac.kr; 3Institute for Veterinary Biomedical Science, College of Veterinary Medicine, Kyungpook National University, Daegu 41566, Republic of Korea; jhj1125@knu.ac.kr (H.J.J.); aam_kim@knu.ac.kr (B.K.); 4Aquatic Disease Control Division, National Fishery Products Quality Management Service, Busan 46083, Republic of Korea; mgkwon@korea.kr (M.-G.K.); sumikim@korea.kr (S.-M.K.)

**Keywords:** *Vp_AHPND_*, strain 19-021-D1, ST 413, O1: non-typeable K, shrimp bioassay

## Abstract

**Simple Summary:**

Acute hepatopancreatic necrosis disease (AHPND) is a plasmid-encoded PirA/B toxin mediated bacterial disease that causes significant economic losses to the global shrimp industry. Since the first AHPND outbreak of *Penaeus vannamei* cultures in Korea in 2016, AHPND-associated *Vibrio parahaemolyticus* (*Vp_AHPND_*) is still considered a major threat to the Korean shrimp industry. Although several diagnostic methods have been developed against AHPND, several concerns have been raised about the misdiagnosis of the pathogen due to the transferability and mutation of the plasmid. In this study, we report the emergence of a natural mutant *Vp_AHPND_* (*pirA*^−^, *pirB*^+^), which was isolated from cultured *P. vannamei* in Korea, and also provide its detailed genomic and pathological characteristics.

**Abstract:**

Acute hepatopancreatic necrosis disease (AHPND) is one of the most important diseases in the global shrimp industry. The emergence of mutant AHPND-associated *V*. *parahaemolyticus* (*Vp_AHPND_*) strains has raised concerns regarding potential misdiagnosis and unforeseen pathogenicity. In this study, we report the first emergence of a type II (*pirA^−^*, *pirB^+^*) natural mutant, *Vp_AHPND_* (strain 20-082A3), isolated from cultured *Penaeus vannamei* in Korea. Phenotypic and genetic analyses revealed a close relationship between the mutant strain 20-082A3 and the virulent Korean *Vp_AHPND_* strain 19-021-D1, which caused an outbreak in 2019. Detailed sequence analysis of AHPND-associated plasmids showed that plasmid pVp_20-082A3B in strain 20-082A3 was almost identical (>99.9%) to that of strain 19-021-D1. Moreover, strains 20-082A3 and 19-021-D1 exhibited the same multilocus sequence type (ST 413) and serotype (O1:Un-typeable K-serogroup), suggesting that the mutant strain is closely related to and may have originated from the virulent strain 19-021-D1. Similar to previous reports on the natural mutant *Vp_AHPND_*, strain 20-082A3 did not induce AHPND-related symptoms or cause mortality in the shrimp bioassay. The emergence of a mutant strain which is almost identical to the virulent *Vp_AHPND_* highlights the need for surveillance of the pathogen prevalent in Korea. Further investigations to elucidate the potential relationship between ST 413 and recent Korean *Vp_AHPND_* isolates are needed.

## 1. Introduction

Acute hepatopancreatic necrosis disease (AHPND, also known as early mortality syndrome, EMS) is one of the most important diseases that can cause sudden and mass mortalities in cultured shrimp worldwide, especially *Penaeus vannamei* and *P. monodon* [1]. The major causative agent of AHPND is *Vibrio parahaemolyticus*, which harbors transferrable pVA1-type plasmids (69–70 kb) that encode two toxin genes (*pirA* and *pirB*) [2,3,4]. Since the first recorded AHPND outbreak in China, the disease has spread to most commercial shrimp-culturing countries and is still causing serious global economic losses to the shrimp aquaculture industry [5]. Moreover, the occurrence of AHPND in Korean shrimp farms was reported in 2016 [1], and the disease is currently listed as a quarantine-requiring crustacean disease that continues to cause serious economic losses in Korea.

*Vibrio parahaemolyticus* producing the PirA/B binary toxins was first reported as the causative agent of AHPND and is commonly referred to as *Vp_AHPND_*. Several reports have revealed transferrable conjugative pVA1-like plasmids encoding *pirA/B* in various *Vibrio* spp., including *V. harveyi* [6], *V. owensii* [7,8], *V. campbellii* [9,10], and *V. punensis* [11]. All exhibit pathogenic potential in shrimp, similar to *Vp_AHPND_*. Furthermore, *pirA/B* genes in pVA1-like plasmids are typically flanked by two identical insertion sequence 5 (IS5) family transposases, IS*Va2* or IS*Val1*, comprising a composite transposon (IS-*pirA/B*-IS) designated Tn*6264* [12] or *pirA/B*-Tn*903* [13]. These ISs or transposons exhibit high genetic variability due to frequent insertion and deletion events involving adjacent genes [14]. These events often lead to mutations in the *pirA/B* region, resulting in the emergence of natural AHPND mutants in the form of the deletion of entire *pirA/B* genes (type I), deletion of entire *pirA* and partial *pirB* genes (type II), and additional *transposase* gene insertion upstream of the *pirA* gene [12,15]. Several studies have confirmed that *Vp_AHPND_* mutants retain pathogenicity in shrimp despite the absence of PirA/B toxin production [16,17].

Although several diagnostic methods have been developed for AHPND, conventional PCR-based *pirA/B* gene detection methods are routinely used in the shrimp culture industry. Concerns have been raised regarding the misdiagnosis of the pathogen, as these results could lead to substantial economic losses in shrimp aquaculture and unforeseen pathogenicity in humans through rapid dissemination [18,19]. Therefore, a detailed investigation is urgently needed into the genetic and pathological characteristics of naturally emerging mutant *Vp_AHPND_* isolates from shrimp farms. 

Since 2017, we have investigated the emergence of *Vp_AHPND_* and other shrimp pathogens in Korean shrimp farms that can cause fatal infections with mass mortality or growth retardation for sustainable aquaculture production. In this study, we report the first emergence of a type II-like mutation in AHPND-associated *V. parahaemolyticus* isolated from cultured *P. vannamei* in Korea, and provide detailed genomic, biological, and pathogenic characteristics to elucidate its potential impact on shrimp culture and public health. These findings will enhance the fundamental understanding of *Vp_AHPND_* mutants and contribute to the development of control and management strategies for this pathogen.

## 2. Materials and Methods

### 2.1. Screening of Mutant Vp_AHPND_ from AHPND-Associated Samples

During the surveillance studies of *Vp_AHPND_* and other shrimp pathogens in Korean shrimp farms between 2020 and 2021, the potential infection (or contamination) of AHPND-associated *Vibrio* was detected by confirming the presence of *pirA* and *pirB* genes using VpPirA-284F/VpPirA-284R and VpPirB-392F/VpPirB-392R primer sets in the duplex PCR assay [2] in various samples, including pond water and its sediments, moribund shrimp and feces, and commercial feed, collected from the shrimp farms that experienced abnormal mortalities. From the PCR-positive samples, the AHPND-associated *Vibrio* isolates were obtained using a standard dilution-plating technique on thiosulfate-citrate-bile salts-sucrose (TCBS; Difco, Detroit, MI, USA) agar plates after incubation at 28 °C for 24 h. Total genomic DNA of the pure colonies was extracted using DNeasy Blood & Tissue Kits (Qiagen, Hilden, Germany) and preliminarily screened for the presence of the pVA1-like plasmids using the 89F/89R primers in the PCR assay as previously described [12]. The presumptive pVA1-like plasmid-harboring colonies were then tested for natural mutant *Vp_AHPND_* isolates by confirming the presence of *pirA* and/or *pirB* genes using a duplex PCR assay [2]. Species of the natural mutant *Vp_AHPND_* isolates were identified by *toxR* gene-sequence analyses [20]. Sequences of the PCR primers used in this study are listed in Table 1. All PCR amplicons were sequenced and compared to representative sequences from other *Vp_AHPND_* strains in the GenBank database using BLAST (www.ncbi.nlm.nih.gov/blast, accessed on 5 March 2024). All the confirmed natural mutant *Vp_AHPND_* isolates were maintained in tryptic soy broth (TSB; Difco), supplemented with 2% NaCl (hereinafter referred to as TSB+) at 28 °C with gentle shaking (150 rpm) for 24 h, and preserved in TSB+ containing 20% glycerol at −80 °C for further analyses.

### 2.2. Identification of Hemolysin Genes in Mutant Vp_AHPND_ by Multiplex PCR

To evaluate the zoonotic potential of the mutant *Vp_AHPND_* isolate in humans, we performed multiplex PCR analysis targeting three hemolysin genes: thermostable direct hemolysin (*tdh*), TDH-related hemolysin (*trh*), and thermolabile hemolysin (*tlh*). Primer sequences used in this study are listed in Table 1. The PCR assays were performed using an initial denaturation at 94 °C for 4 min, followed by 30 cycles of denaturation at 94 °C for 1 min, annealing at 50 °C for 1 min, and extension at 72 °C for 1 min, with a final extension step at 72 °C for 10 min. Subsequently, representative PCR amplicons were sequenced and compared with the respective reference sequences in the GenBank database using BLAST (www.ncbi.nlm.nih.gov/blast, accessed on 5 March 2024). A previously well-characterized virulent Korean *Vp_AHPND_* isolate, strain 19-021-D1 [1], was analyzed under identical conditions for comparative assessment.

### 2.3. Assessment of Hemolytic Activity in Mutant Vp_AHPND_ Using Kanagawa Phenomenon (KP) Test

The total hemolytic activity of the mutant *Vp_AHPND_* isolate was analyzed in vitro using the Kanagawa phenomenon (KP) test following a modified protocol, previously described [23]. KP blood agar plates were prepared using TSA supplemented with 5% (*v*/*v*) defibrinated sheep blood and 7% (*w*/*v*) NaCl. Bacterial colonies, cultured overnight on TSA+ plates at 37 °C, were streaked onto blood agar plates and further incubated at 37 °C for 24 h. The presence of transparent hemolysis zones surrounding the bacterial colonies on blood agar, indicative of β-hemolysis was interpreted as KP-positive. Greenish discoloration around colonies, indicative of α-hemolysis, or no reaction in the surrounding medium, indicative of γ-hemolysis, was considered KP-negative. The total hemolytic capability of the Korean *Vp_AHPND_* strain 19-021-D1was also evaluated under the same conditions for comparative analysis.

### 2.4. Antimicrobial Susceptibility Test of Mutant Vp_AHPND_

The antimicrobial susceptibility of the mutant *Vp_AHPND_* isolate was evaluated using the disk-diffusion method [24] according to the guidelines of the Clinical and Laboratory Standards Institute (CLSI) [25]. A total of 15 antimicrobial agents (Oxoid Ltd., Basingstoke, UK), including amikacin (AK, 30 µg), ampicillin (AMP, 10 µg), ampicillin-sulbactam (SAM, 20 µg), azithromycin (AZM, 15 µg), cefepime (FEP, 30 µg), cefotaxime (CTX, 30 µg), cefoxitin (CFX, 30 µg), ceftazidime (CAZ, 30 µg), chloramphenicol (C, 30 µg), ciprofloxacin (CIP, 5 µg), doxycycline (DXT, 30 µg), levofloxacin (LEV, 5 µg), piperacillin (PRL, 100 µg), tetracycline (TE, 30 µg), and trimethoprim/sulfamethoxazole (SXT, 1.25/23.75 µg) were used for the disk-diffusion test. Bacterial colonies grown overnight on TSA+ were suspended in 3% NaCl (*w*/*v*) saline solution and adjusted to a McFarland standard of 0.5. Afterwards, 100 µL of the bacterial suspension was spread on Mueller–Hinton agar (HiMedia, Mumbai, India) plates containing 3% NaCl (*w*/*v*) and incubated at 37 °C for 18 h [26]. The diameters of the transparent inhibitory zones (mm) were then measured and interpreted based on the CLSI M45 guidelines [25]. However, as the criteria for doxycycline and azithromycin are not available in the CLSI M45 guidelines, these two antibiotics were interpreted based on CLSI M100 standards [27,28]. Additionally, multiple antibiotic resistance (MAR) phenotype (MARP) patterns and MAR indices were analyzed. The MAR index was calculated as the ratio of the number of antimicrobials to which bacteria showed resistance to the number of antimicrobials used in this study.

### 2.5. Whole Genome Sequencing of Mutant Vp_AHPND_

Total genomic DNA of the mutant *Vp_AHPND_* isolate was extracted using the DNeasy Blood & Tissue Kit (Qiagen, Hilden, Germany) and sequenced using a hybrid approach: (i) using the PacBio RSII System (Pacific Biosciences, Menlo Park, CA, USA) to construct a 20-kb SMRTbell^TM^ template library, and (ii) using the HiSeq X-10 platform (Illumina, San Diego, CA, USA) to prepare a DNA library using the TruSeq Nano DNA Library Prep Kit (Illumina). PacBio reads (1,271,599,957 bp; 121,815 reads) were de novo assembled using HGAP (v.3.0) [29]. The filtered Illumina paired-end reads (912,987,978 bp; 6,046,278 reads) were mapped to the PacBio-assembled genome using BWA-MEM v.0.7.17 [30], and the final error correction was performed using Pilon v.1.21 [31]. Genome annotation was performed using the National Center for Biotechnology Information prokaryotic genome annotation pipeline (http://www.ncbi.nlm.nih.gov/books/NBK174280/, accessed on 13 April 2024). The complete genome was visualized using the Pathosystems Resource Integration Center (PATRIC) server v.3.6.12 [32]. 

Using the sequenced genome, we identified the serotype of the mutant isolate by comparing the gene clusters for lipopolysaccharide (LPS) and capsular polysaccharide (CPS), which determine the O-serogroup and K-serogroup, respectively, against the reference O and K serogroups in the VPsero [33] and *Kaptive* database [34]. Multilocus sequence typing (MLST) was performed to determine the sequence type (ST) of mutant *Vp_AHPND_*, as previously described [35]. A total of seven housekeeping genes (*dnaE*, *gyrB*, *recA*, *dtdS*, *pntA*, *pyrC*, and *tnaA*) of the sequenced genome were screened and compared in silico with those available in the PubMLST database (https://pubmlst.org/organisms/vibrio-spp, accessed on 25 April 2024), and its respective allele numbers and ST were generated. To evaluate the pathogenic potential of the mutant *Vp_AHPND_*, putative virulence- and AMR-associated genes and mobile genetic elements (MGEs), including integrative and conjugative elements (ICEs), were screened in silico using VFDB (http://www.mgc.ac.cn/VFs/. accessed on 26 April 2024), CARD (https://card.mcmaster.ca, accessed on 26 April 2024), and VRprofile2 (https://tool2-mml.sjtu.edu.cn/VRprofile, accessed on 26 April 2024), respectively. Additionally, the integrated prophage regions in the genome were identified using PHASTEST web server (https://phastest.ca/, accessed on 1 May 2024).

### 2.6. Genetic Characterization of pVA-1-like Plasmid in Mutant Vp_AHPND_

For detailed genomic characterization of the AHPND-associated plasmid in the mutant *Vp_AHPND_*, we first determined the genotype of the pVA-1-like plasmid in the mutant *Vp_AHPND_* by PCR [1] and direct sequence comparison with other virulent AHPND plasmids available in the GenBank database. The orthologous average nucleotide identity (OrthoANI) algorithm [36] was used to assess the overall pVA-1-like plasmid genome similarity between the Korean mutant *Vp_AHPND_* isolate and other *Vp_AHPND_* strains of different geographical origins. Finally, the Tn6264 transposon region that generally contains four major open reading frames (ORFs)/genes (arranged in the order IS (inverted)-*pirA*-*pirB*-IS (direct)) in the pVA-1-like plasmid were manually searched and compared to those regions from plasmid pVp_Kor-D1-2 (CP046414) and plasmid pVPA3-1 (KM067908), respectively.

### 2.7. Shrimp Bioassays

Animal use and experimental protocols were reviewed and approved by the Animal Research Ethics Committee of Kyungpook National University (KNU 2021-0066, April 2021). Pacific white shrimp (*P. vannamei*) at the juvenile stage (approximately 0.2 g) were purchased from a local shrimp farm (Jeju Province, Republic of Korea) and transported to the Laboratory of Aquatic Biomedicine, College of Veterinary Medicine, Kyungpook National University (Daegu, Republic of Korea). The shrimp were reared to an average weight of 0.5 ± 0.05 g in 700-L maintenance tanks filled with aerated artificial seawater (25–28 °C, 25 ppt salinity). Before the bioassay, three shrimps were randomly selected and examined to confirm that the tested shrimp (hepatopancreas, HP) did not carry *pirA* and/or *pirB* genes using a duplex PCR assay, as described previously [2].

To determine the potential pathogenicity of the Korean mutant *Vp_AHPND_* isolate, shrimp bioassays were conducted using the immersion method described by [12,37], with two replicate tanks each for the mutant *Vp_AHPND_* isolate and virulent *Vp_AHPND_* strains. Briefly, a total of two tanks (working water volume: 15 L) were filled with artificial seawater (salinity, 25 ppt; temperature, 25–28 °C), and healthy Pacific white shrimps (n = 5; weight: 0.4–0.5 g) were stocked in each tank. For infection, the mutant *Vp_AHPND_* isolate was cultured in TSB+ at 28 °C with gentle shaking (150 rpm) for 24 h until it reached a concentration of approximately 10^9^ CFU/mL. Shrimps were immersed in water containing bacteria at a concentration of approximately 10^6^ CFU/mL, following the protocol for the detection of mutant *Vp_AHPND_* [12], and monitored until the termination day (day 7). As a negative control, two tanks with healthy shrimp (n = 5 each) were maintained without bacteria. As a positive control, two tanks with healthy shrimp (n = 5 each) were exposed to the virulent *Vp_AHPND_* strain, 13-028/A3 (Vietnam, 2013) [37]. Experimental shrimp were fed with commercial shrimp feed at 5% of shrimp biomass, and survival was monitored and recorded for 7 days. For AHPND confirmation, representative (live or moribund) shrimp samples were fixed in 95% ethanol for PCR assays, and fixed in Davidson’s AFA (alcohol, formaldehyde, acetic acid) fixative for histopathology analysis, at 24 h post-infection from each group.

### 2.8. Nucleotide Sequence Accession Numbers

The complete genome sequence of the mutant *Vp_AHPND_* strain 20-082A3 was deposited in the GenBank database under the accession numbers NZ_CP065369.1 (chromosome 1), NZ_CP065370.1 (chromosome 2), NZ_CP065371.1 (pVp-20-082A3A), and NZ_CP065372.1 (pVp-20-082A3B).

## 3. Results

### 3.1. Natural Mutant Vp_AHPND_ Strain 20-082A3 from AHPND-Associated Shrimp Samples

During the surveillance study, several presumptive AHPND-associated samples that may have contained *Vp_AHPND_* were detected using PCR assays. From these samples, we attempted to isolate a natural mutant strain of AHPND-associated *Vibrio* using the TCBS agar plating method. All greenish presumptive *Vibrio* colonies were sampled and assessed by PCR to confirm the presence of natural mutant *Vp_AHPND_* isolates. We confirmed the presence of two types of mutant *Vp_AHPND_* colonies along with typical *Vp_AHPND_* colonies: (i) PCR-positive for the pVA1-like plasmid, but negative for *pirA* and *pirB*; and (ii) PCR-positive for the pVA1-like plasmid and *pirB*, but negative for *pirA*. From these samples, we attempted to isolate a natural mutant strain of AHPND-associated *Vibrio* using the TCBS agar plating method. And finally, we successfully obtained a natural mutant *Vp_AHPND_* isolate, designated as strain 20-082A3, from the HP of *P. vannamei*, cultured in a private shrimp farm located in Boryung-si (Chungcheongnam-do, Korea), and collected in 2020 (Table 2). This isolate was PCR-positive for the pVA1-like plasmid and *pirB*, but negative for *pirA*. Based on sequencing analyses of the obtained PCR amplicons of the pVA1-like plasmids, *pirB*, and *toxR* genes, the strain was confirmed to encode pVA1-like plasmids and *pirB*, and was finally identified as *V. parahaemolyticus*.

### 3.2. Hemolytic and AMR Features of Mutant Vp_AHPND_ Strain 20-082A3

Multiplex PCR for hemolysin-associated gene detection revealed that two Korean-originated AHPND-associated *V. parahaemolyticus* strains (19-021-D1and 20-082A3) were positive for the *tlh* gene, but negative for *trh* and *tdh* genes. Consistent with these results, the two strains exhibited greenish discoloration around bacterial colonies on 7% sheep blood agar, indicating α-hemolysis (KP-negative) (Table 2). 

The antimicrobial susceptibility test confirmed that the two AHPND-associated Korean *V. parahaemolyticus* strains were commonly resistant to ampicillin (10 μg); however, strain 20-082A3 exhibited resistance to a broader range of antimicrobials showing different MARP patterns compared to the 19-021-D1 strain. The mutant strain showed resistance to piperacillin (100 μg, β-lactam) and ampicillin-sulbactam (20 μg, β-lactamase inhibitor), but was susceptible to amikacin (30 μg, aminoglycoside). Moreover, the MAR index of strain 20-082A3 was estimated to be 0.2, which was higher than the MAR index of 0.13 observed for strain 19-021-D1(Table 2 and Figure 1).

### 3.3. General Features of Mutant Vp_AHPND_ Strain 20-082A3 Genome

The sequenced genome of the mutant *Vp_AHPND_* strain 20-082A3 consists of two circular chromosomes and two plasmids (pVp_20-082A3A and pVp_20-082A3B), which are 5,601,613 bp long with a GC content of 45.3% and contain 4986 protein-coding genes (CDSs). The annotated CDSs in the total genome were visualized using a genome map and categorized based on their functions (Appendix A). Chromosome 1 measured 3,610,915 bp in length (183 × coverage) with a G + C content of 45.1%, and chromosome 2 was 1,802,312 bp long (168 × coverage) with a G + C content of 45.7%. The plasmid pVp_20-082A3A was 121,293 bp in length (337 × coverage) with a G + C content of 44.1%. Finally, the pVA1-type plasmid pVp_20-082A3B was 67,093 bp long (262 × coverage), with a G + C content of 46.1% (Table 2).

Using the sequenced genome, we first determined the serotype of the mutant *Vp_AHPND_* strain 20-082A3 by in silico analyses: (i) VPsero predicted the entire LPS locus in the mutant isolate as belonging to the O1-serogroup; and (ii) the CPS locus of strain 20-082A3 was not identical to those of the other reported K serogroups. The locus was most similar to K-locus 105, which was reported from the *Vp_AHPND_* strain M0605 isolated from AHPND/EMS-affected shrimp in Mexico [38], with the K-serogroup unknown by the *Kaptive* web-based tool. Therefore, the serotype of strain 20-082A3 was finally determined to be O1:KUT (untypeable K serogroup). Interestingly, the O- and K-serogroup-determining regions were 100% identical to those of the previously reported Korean *Vp_AHPND_* strain 19-021-D1. Moreover, we performed in silico MLST analyses to determine the ST of strain 20-082A3. Its allele profile was confirmed to be 47-8-166-19-28-46-121, and its ST was finally assigned to 413, which was also identical to that of strain 19-021-D1 [35]. 

A total of 34 MGEs (23 on chromosome 1, six on chromosome 2, four on plasmid pVp_20-082A3A, and one on pVp_20-082A3B) were detected using VRprofile2 analysis (Appendix A). Notably, chromosome 1 possessed nine virulence-associated genes spanning three MGEs: Type 3 secretion system (T3SS) family genes, *vecA*, *vopS*, *vopR,* and *vopQ* were located within a prophage region, while *vpadF* was identified in the ICE and *ugd*, *cap8E*, *rfaD,* and *wbtL* were found in other ICE regions. Additionally, the presence of Type 6 secretion systems (T6SS1 and T6SS2) and genomic Islands (GIs) in strain 20-082A3 was further investigated by manual comparison using the genome of a well-studied clinical *V*. *parahaemolyticus* strain RIMD 2210633 (NC_004603.1) [39,40]. From the genome of the two Korean isolates, the T6SS1 and T6SS2 were respectively identified on chromosomes 1 and 2, with high sequence identity (>98%) to the strain RIMD 2210633 (Appendix A). Furthermore, a total of five GIs (VPaI-1, VPaI-2, VPaI-3, VPaI-6, and VPaI-7) were detected on chromosome 1 of the two strains, with high sequence identity (>98%) to those in strain RIMD 2210633 (Appendix A). The toxin-related genes on the chromosome of mutant 20-082A3 were also compared with those of strain 19-021-D1 using VFDB and showed relatively similar results (Appendix A).

Additionally, the web-based tool PHASTEST predicted four prophage regions across the two chromosomes. Among these regions, region I was marked as incomplete (score < 70), whereas the other regions (II, III, and IV) were identified as intact (score > 90) (Appendix A). Based on the PHASTEST database, each region showed significant similarity to proteins from specific phages: Region I with *Lactobacillus* phage AQ113 (NC_019782), Region II with *Burkholderia* phage BcepMu (NC_005882), Region III with *Escherichia* phage D108 (NC_013594), and Region IV with *Vibrio* phage VEJphi (NC_012757). No prophage region was detected in the plasmid. Detailed genomic analyses provided insights into the specific features of three intact prophage regions. Region II, spanning 24.1 kb (973,220–997,337 bp, 46.8% G + C content) on chromosome 1, encompasses 34 CDSs and prominently functional proteins associated with phage structure and nucleotide metabolism. On chromosome 2, Region III of 25.2 kb (698,613–723,883 bp, 47.1% G + C content) encodes 39 CDSs, including a putative endolysin protein (CDS 9) similar to that of *Acinetobacter* phage phiAC-1 (NC_028995). Similarly, Region IV of 12.1 kb (1,781,378–1,793,551 bp, 43.5% G + C content) on chromosome 2 comprises 19 CDSs, predominantly identified as phage head proteins similar to those of *Vibrio* phages.

CARD analyses detected several ARGs in the following locations: (i) Chromosome 1: Two antibiotic efflux pump genes, *txR* (tetracycline resistance) and *crp* (macrolide, fluoroquinolone, and penam resistance), along with two antibiotic target alteration genes, *parE* (fluoroquinolone resistance) and *vanT* (glycopeptide resistance); (ii) Chromosome 2: Two antibiotic efflux pump genes, *adeF* (fluoroquinolone and tetracycline resistance) and an antibiotic inactivation gene, *bla_CARB-18_* (penam resistance). Additionally, VRprofile2 identified two ARGs, including *tet(34)* (tetracycline resistance) located on chromosome 1. No ARGs were detected in any plasmids (Appendix A). 

### 3.4. Detailed Features of pVA1-like Plasmid in Avirulent Mutant Strain 20-082A3

During the genotyping of plasmid pVp_20-082A3B, a PCR amplicon of 482 bp was obtained, and the mutant *Vp_AHPND_* strain 20-082A3 was confirmed to be Asian-type. The sequenced genome of plasmid pVp_20-082A3B (67,093 bp) encoded 75 CDSs, indicating that it has a shorter genome length and contains fewer protein-coding genes than other AHPND-associated plasmids, including the pVp_Kor-D1-2 plasmid (68,848 bp, 81 CDSs, CP046414.1) from strain 19-021-D1 (Table 3). OrthoANI values were analyzed to compare the plasmid pVp_20-082A3B with other available AHPND-associated plasmids from different geographical origins, and the highest OrthoANI value was obtained for the Korean AHPND plasmid pVp_Kor-D1-2 (CP046414.1, 99.9%) (Figure 2). As expected, the Tn3-like transposon that was identified in Mexican-type *Vp_AHPND_* isolates [12] was not found in plasmid pVp_20-082A3B.
Moreover, a total of four short sequence repeats (SSRs) consisting of 9 bp sequences (5′-TTGTTTTTC-3′) [1] were found in the plasmid (region: 3385–3420). However, the Tn6264 transposon region that generally contains the four major ORFs/genes (arranged in an order of IS (inverted)—*pirA—pirB*—IS (direct)) in plasmid pVp_20-082A3B differed from the plasmid pVp_Kor-D1-2 as follows:
(i) Although the IS in both ends were 100% identical to the other AHPND-associated plasmids, the total length of Tn6264 transposon region of plasmid pVp_20-082A3B was 3677 bp, whereas its length of plasmid pVp_Kor-D1-2 was 5427 bp; (ii) from the left 18-bp inverted repeat (IR: GAATTACGCAACAAAGCC) [1] located at the region (16,930–16,913) of plasmid pVp_20-082A3B, the entire *pirA* gene and its upstream hypothetical protein-containing region were deleted; (iii) the deletion continued to the *pirB* gene and a total of 305 bp deletion occurred from the starting codon of the gene (Figure 3).

### 3.5. Shrimp Bioassays Using Mutant Vp_AHPND_ Strain 20-082A3

A shrimp bioassay was performed to determine the pathogenicity of the Korean mutant *Vp_AHPND_* isolate. The results of the shrimp laboratory bioassays were similar to those previously reported from other mutant *Vp_AHPND_* strains [12,15]. As with the shrimps in the negative control (without strain 20-082A3), the survival rate of shrimps challenged by the mutant *Vp_AHPND_* isolate was 100%, and the live shrimp showed normal external appearances and feeding behaviors until the termination day of the experiment (day 7). In contrast, the shrimp in the positive control (challenged by the virulent *Vp_AHPND_* strain, 13-028/A3) showed typical AHPND symptoms and died at 48–72 h.

PCR analysis of the shrimp confirmed the presence of the toxin genes (*pirA* and *pirB*) in the shrimp HP of the positive control, whereas only the *pirB* gene was detected in the shrimp HP challenged by the mutant *Vp_AHPND_* isolate (20-082A3), and the toxin genes were not detected in the shrimp HP of the negative control at 24 h (Figure 4). Although the natural mutant *Vp_AHPND_* strain 20-082A3 did not cause any cumulative mortalities to the indicator *P. vannamei*, histopathological examination of the surviving shrimp showed bacterial colonization of the HP with light atrophy, which differed from those observed in the positive control group (Figure 5).

## 4. Discussion

In this study, we reported the first emergence of the mutant *Vp_AHPND_* strain 20-082A3 from culturing *P. vannamei* in Korea, along with its genomic and pathological characteristics. Based on a detailed genetic investigation of the AHPND-associated plasmid in mutant strain 20-082A3, we confirmed that the newly isolated natural mutant strain possessed the Asian-type pVA1-like plasmid with a 1750 bp deletion that possesses the entire *pirA* and partial *pirB* genes. Previously, we classified *Vp_AHPND_* mutants into two types based on the deletion region on the pVA1 plasmid: type I (with loss of the entire *pirA/B* gene and upstream IS*Val1*), and type II (with loss of the entire *pirA* and partial *pirB* genes), which did not induce pathogenicity in shrimp with AHPND lesions [12]. Additionally, a more recent study proposed a new class of mutant type III, characterized by the presence of *pirA/B* binary genes, but lacking PirA/B binary toxin production and pathogenicity [15]. The type III mutant strain R14 failed to translate the mRNA corresponding to the *pirA/B* genes and did not induce the typical histopathological appearance of AHPND in shrimp HP. According to the previously reported classification model, the newly-isolated natural mutant strain 20-082A3 was determined to be a type II mutant, which is characterized by the largest number of nucleotide deletions (c.a., 1.7-kbp) that includes the complete *pirA* and partial *pirB* genes reported to date. Although another unclassified mutant *Vp_AHPND_* strain XN87, which showed a frameshift mutation resulting in no production of PirA/B toxin, exhibited weak mortality without AHPND lesions in shrimp [16], strain 20-082A3-challenged shrimp in this study did not exhibit typical AHPND lesions or mortality similar to the other mutant *Vp_AHPND_* strains mentioned above [12,15]. These results could be explained as follows: (i) the type II mutant strain 20-082A3 lost the nucleotide region encoding the PirA subunit; and (ii) it possessed a total of 305 bp deletion from the starting codon of the *pirB* gene, and also lost the upstream region of *pirA^VP^*, including a promoter for *pirA/B* operon, leading to no expression of the remaining PirB subunit. Therefore, the bacterial colonization of HP with light atrophy observed in this study could have resulted from other virulent factors rather than the production of an incomplete PirB subunit. 

With great interest, the AHPND-associated plasmid pVp_20-082A3B in the strain 20-082A3 was confirmed to be very closely related to the plasmid pVp_Kor-D1-2 in the virulent strain 19-021-D1 [1]. First, a direct sequence comparison of the two-plasmid genome revealed the highest similarity (>99.9%) during the OrthoANI analysis along with the other available AHPND-associated plasmids in the GenBank database. Second, the Tn3-like transposon identified in Mexican-type *Vp_AHPND_* isolates was not detected in both plasmids. Third, the same number of SSRs was detected in the plasmids. We also evaluated the potential relatedness of the AHPND-associated plasmid-harboring strains 20-082A3 and 19-021-D1. As we expected, the two *V. parahaemolyticus* strains exhibited very similar phenotypical and genetic features. First, the two strains showed the same patterns of the hemolysin-associated genes (*tlh*^+^, *trh*^−^, *tdh*^−^) and α-hemolytic nature by KP test. Second, the in silico serotyping analyses confirmed that the O- and K-serogroup determining regions of the two strains were 100% identical and finally determined to be O1:KUT. Third, the in silico MLST analyses also revealed that the two strains were identical in the allele profile (47-8-166-19-28-46-121) and the ST (413). These results strongly indicate that the natural mutant *Vp_AHPND_* strain 20-082A3 is closely related and might have originated and evolved from the virulent Korean *Vp_AHPND_* strain 19-021-D1. Recent research on the MLST ST of *Vp_AHPND_* isolates from different geographical origins revealed that ST 413 clusters were only found in Thailand, Korea, and Vietnam, suggesting a transmission route between these countries [41]. This study confirmed that a specific ST of *Vp_AHPND_* could be prevalent in a geographically adjacent region of Vietnam. Based on these findings, it can be speculated that the strain 20-082A3-isolated shrimp culture farm might have been affected by the *Vp_AHPND_* ST 413 cluster, which caused an outbreak in Taean-gun (Chungcheongnam-do) in 2019 and evolved from the virulent Korean *Vp_AHPND_* strain 19-021-D1 (Appendix A). This hypothesis can be supported by the comparison of the antimicrobial susceptibility of the two strains; strain 20-082A3 (MAR index 0.2) showed a broader antibiotic resistance pattern than the virulent strain 19-021-D1(MAR index 0.16). In general, an MAR index > 0.2 suggests that the bacteria could have originated from a high-risk contaminant source using multiple antibiotics [42]. These results potentially suggest that the Korean mutant *Vp_AHPND_* strain has enhanced antibiotics resistance, allowing it to survive in multi-antibiotic environments and might subsequently develop resistance to additional antibiotics in the future. The differences in the resistance patterns and MAR indices between strains 20-082A3 and 19-021-D1 underscore the importance of monitoring isolates in shrimp aquaculture for diverse resistance mechanisms. 

*V. parahaemolyticus* is a zoonotic pathogen that infects shrimp and also causes diarrhea or gastroenteritis in humans upon the consumption of raw or undercooked seafood [43]. Although its virulence mechanisms in humans have not been fully elucidated, T3SS and hemolysins play significant roles in its pathogenicity [44]. Based on whole genome analyses, the mutant strain encoded three well-known effectors (*vopS*, *vopR*, and *vopQ*) [45,46,47] for T3SS-induced cell death, the T3SS-specific chaperone *vecA* on the prophage region, *vpadF* [7] on the ICE region, and T6SS1/T6SS2 along with five GIs, thus suggesting its potential pathogenicity in humans. Additionally, the PHASTEST analysis identified three intact prophage regions (score > 90%), suggesting their recent incorporation into the bacterial genome [48]. These results indicate that the mutant strain 20-082A3 might potentially disseminate these host cell death-related virulence factors globally through MGEs, thereby contributing to their widespread distribution and genetic diversity. Therefore, continuous monitoring and surveillance of *Vp_AHPND_* focusing on the acquisition of virulence genes or ARGs will be needed to mitigate the potential impacts on the shrimp culture industry and public health. Further genome-based investigations to elucidate the potential relationship between ST 413 and recent Korean *Vp_AHPND_* isolates are currently in progress.

## 5. Conclusions

This study investigated the detailed genomic and pathological characteristics of the firstly emerged natural mutant *Vp_AHPND_*, isolated from cultured *Penaeus vannamei* in Korea. Based on the detailed AHPND-associated plasmid sequence analyses, it was found that the natural mutant *Vp_AHPND_* strain 20-082A3 harbored pVA1-type plasmid (pVp_20-082A3B) with deletion of entire *pirA* and partial *pirB* genes, and thus confirmed it to be type II mutant *Vp_AHPND_*. Further genetic analyses revealed that the mutant strain 20-082A3 and the virulent Korean *Vp_AHPND_* strain 19-021-D1, which caused an outbreak in 2019, exhibited the same MLST (ST 413) and serotype (O1:KUT), suggesting that the mutant strain is closely related to and may have originated from the virulent strain 19-021-D1. The emergence of a mutant strain which is almost identical to the virulent *Vp_AHPND_* highlights the need for surveillance of the pathogen prevalent in Korea and further investigation to elucidate the potential relationship between ST 413 and recent Korean *Vp_AHPND_* isolates will be needed.

## Figures and Tables

**Figure 1 animals-14-02788-f001:**
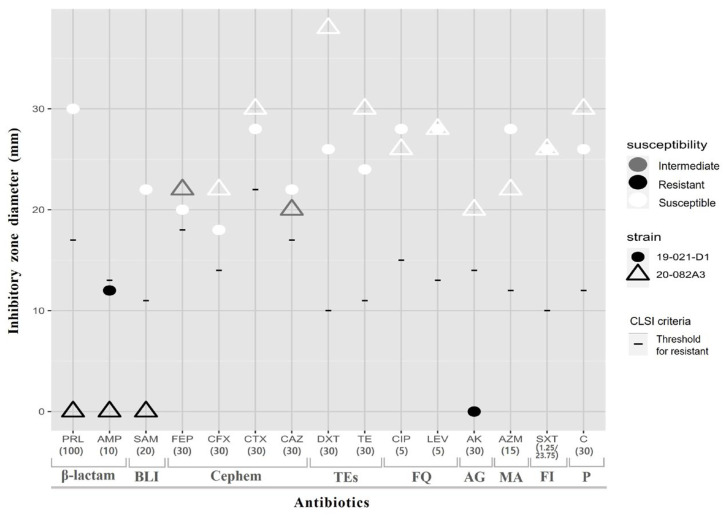
Comparison of antimicrobial susceptibility between *Vp_AHPND_* strain 19-021-D1 and mutant strain 20-082A3. The antimicrobial phenotype was determined using the antibiotic disks and selected according to the CLSI M45 recommendation. The X-axis represents the antibiotics tested in this study, while the Y-axis depicts the inhibition zone diameter (mm). Antibiotic classes are marked below the X-axis using square brackets. Susceptibility to antibiotics is represented as follows: black (resistant), grey (intermediate), and white (susceptible). Dot shapes indicate the isolates and CLSI criteria: circles for strain 19-021-D1, triangles for strain 20-082A3, and lines representing the inhibition zone diameter threshold for resistance. Abbreviations for antibiotic classes are as follows: BLI (B-lactamase inhibitor), TEs (tetracyclines), FQ (fluoroquinolones), AG (aminoglycosides), MA (macrolides), FI (folate pathway inhibitors), and P (phenicol).

**Figure 2 animals-14-02788-f002:**
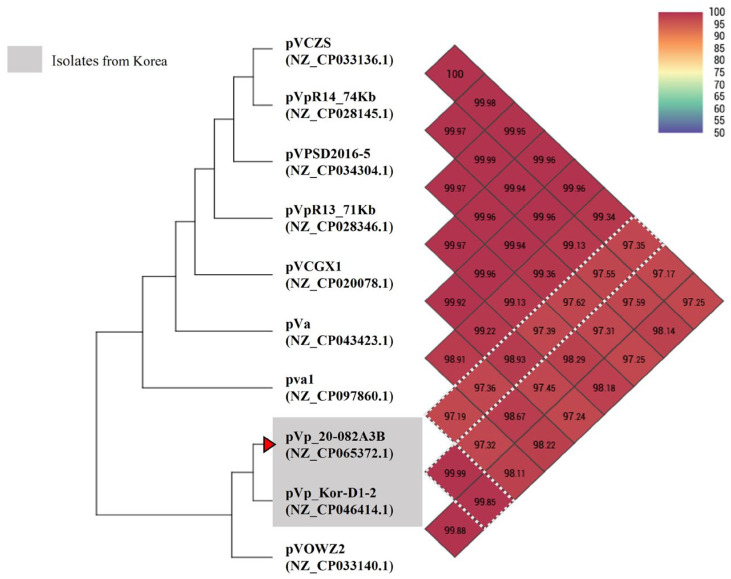
Heatmap displaying the average nucleotide identity by OrthoANI between AHPND-associated plasmids from various geographical locations, including the mutant plasmid pVp_20-082A3B. On the taxon, a grey box represents the pVA1-type plasmids from Korea, while a red triangle marks the mutant pVA1-type plasmid isolated in this study. White dashed lines on the heatmap highlight its OrthoANI values with other AHPND-associated plasmids, providing a clear comparison of their genetic similarity. Heatmap was generated using the Orthologous Average Nucleotide Identity Tool (OAT) available at EZBioCloud (https://www.ezbiocloud.net/tools/orthoani, accessed on 25 April 2024).

**Figure 3 animals-14-02788-f003:**
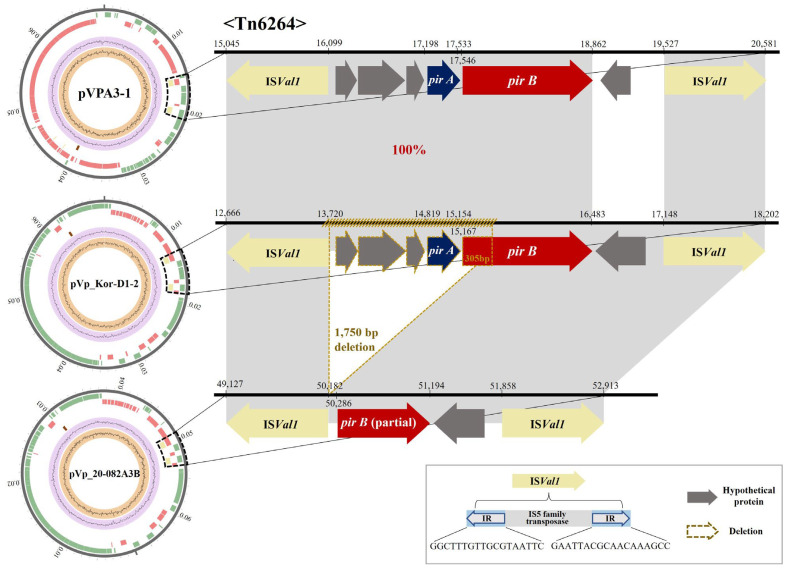
Comparative genomic analysis of the Tn6264 regions of the three pVA1-like plasmids: pVPA3-1 (Vietnam), pVp_Kor-D1-2 (Korea), and the mutant plasmid pVp_20-082A3B (Korea). The genome map illustrates the differences in the arrangement of CDSs within the Tn6264 region of each plasmid. The Tn6264 region is marked as black dashed lines on the circular plasmid maps and zoomed-in as horizontal black bars on the right side. CDSs within the Tn6264 region are represented as colored horizontal arrows, indicating their presumptive function: blue for PirA, red for PirB, dark grey for hypothetical proteins, and yellow for insertion sequence IS*Val1*. The direction of the arrows indicates the transcription orientation of the CDSs. The structure of IS*Val1*, flanking the ends of the Tn6264 region, is shown in the bottom box. Homologous regions with 100% nucleotide identity, according to BLASTn analysis, are connected by light grey blocks between the sequence bars. The deleted region and CDSs in the mutant plasmid pVp_20-082A3B, compared to the reference plasmid pVp_Kor-D1-2, are highlighted by a dark yellow dashed line in the bottom box. For a comprehensive description of the tracks used in the genome map, refer to Appendix A.

**Figure 4 animals-14-02788-f004:**
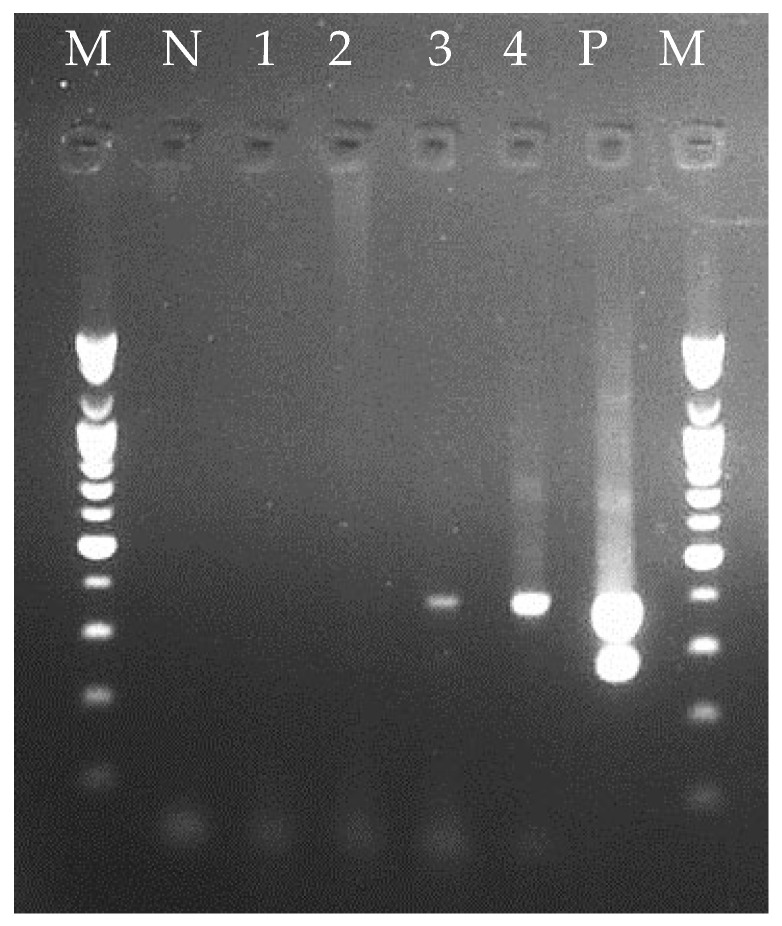
A duplex PCR assay was conducted to confirm the presence of the toxin genes (*pirA* and *pirB*). Lane M: 100-bp DNA ladder; lane N: negative control; lane 1: shrimp were randomly selected and confirmed negative for toxins before bioassay; lane 2: shrimp from negative control (without the mutant strain 20-082A3); lanes 3 and 4: shrimp challenged by the mutant strain 20-082A3; lane P: shrimp from positive control at 24 h after infection.

**Figure 5 animals-14-02788-f005:**
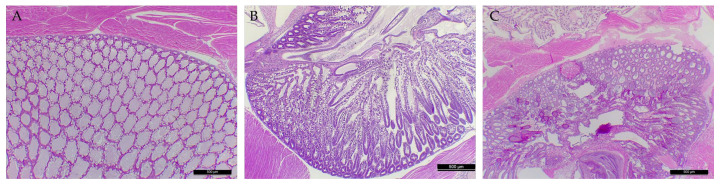
Histopathology examination using Davidson’s AFA-fixed tissue sections from experimental shrimp. The HP of the negative control (without the mutant strain 20-082A3) had a normal appearance with normal cell functions (**A**), and the HP of surviving shrimps challenged by the mutant strain 20-082A3 showed bacterial colonization of the HP with light atrophy (**B**). However, histopathological signs in the positive control (*Vp_AHPND_* strain 13-028/A3) showed massive bacterial infection-related sloughing of HP tubule epithelial cells and hemocytic inflammation (**C**).

**Table 1 animals-14-02788-t001:** Detailed information on the PCR primers used in this study.

Diagnosis	Target Gene	Primer Name	Primer Sequence (5′–3′)	Amplicon Size (bp)	Ref.
AHPND toxin gene detection	*pirA* (duplex)	VpPirA-284F	TGACTATTCTCACGATTGGACTG	284	[2]
		VpPirA-284R	CACGACTAGCGCCATTGTTA		
	*pirB* (duplex)	VpPirB-392F	TGATGAAGTGATGGGTGCTC	392	
		VpPirB-392R	TGTAAGCGCCGTTTAACTCA		
AHPND plasmid detection	pVA1-like plasmid	89F	GTCGCTACTGTCTAGCTGAA	490	[12]
		89R	ACGGCAAGACTTAGTGTACC		
	Plasmid genotype	MX-345F	TACCAGCTCTAACAAGGCCA	345	[1]
		MX-345R	AACGTTCCAAGGAGTCGAGT		
		Asia-482F	TGAACCGTTCCTCATGCTCT	482	
		Asia-482R	TCAAAGCAGCCCAGACAAAC		
*V. parahaemolyticus* identification *	*toxR*	toxR-F	GTCTTCTGACGCAATCGTTG	368	[20]
		toxR-R	ATACGAGTGGTTGCTGTCATG		
	*tdh*	tdh-F	CCATTCTGGCAAAGTTATT	534	[21]
		tdh-R	TTCATATGCTTCTACATTAAC		
	*trh*	*trh*-F	TTGGCTTCGATATTTCAGTATCT	500	[22]
		*trh*-R	CATAACAAACATATGCCCATTTCCG		
	*tlh*	*tlh*-F	AGCGGATTATGCAGAAGCAC	150	[21]
		*tlh*-R	ATCTCAAGCACTTTCGCACG		

* *toxR*, Toxin operon; *tdh*, Thermostable direct hemolysin; *trh*, TDH-related hemolysin; *tlh*, Thermolabile hemolysin.

**Table 2 animals-14-02788-t002:** Detailed information of the virulent *Vp_AHPND_* strain 19-021-D1 and mutant strain 20-082A3.

Features	*V*. *parahaemolyticus*	
	19-021-D1	20-082A3
Isolate information		
Isolation year	2019	2020
Isolation site	Taean-gun/Chungcheongnam-do	Boryeong-si/Chungcheongnam-do
Isolation source	Shrimp pond water	*P*. *vannamei*
Antibiotic resistance pattern	AMP, AK	PRL, AMP, SAM
MAR index *	0.13	0.20
Hemolysin gene (*tlh*, *trh*, *tdh*)	+, −, −	+, −, −
Kanagawa phenomenon	α-hemolysis	α-hemolysis
Genomic information		
Genome size (bp)	5,579,380	5,601,613
Chromosome 1	3,577,848	3,610,915
Chromosome 2	1,814,246	1,802,312
Plasmid 1	(pVp_Kor-D1-1) 118,438	(pVp_20-082A3A) 121,293
Plasmid 2	(pVp_Kor-D1-2) 68,848	(pVp_20-082A3B) 67,093
G + C content (%)	45.1	45.3
Total genes	5297	5226
Protein-coding genes (CDS)	5052	4986
rRNAs	37	37
tRNAs	133	132
ncRNAs	4	4
Pseudogenes	71	67
Serotype	O1:KUT ***	O1:KUT
MLST sequence type **	413	413

* MAR index represents the number of antibiotics to which bacteria showed resistance divided by the total number of antibiotics used. ** MLST, Multilocus sequence type. *** KUT, Un-typeable K-serogroup.

**Table 3 animals-14-02788-t003:** Detailed features on the pVA1-like plasmids in the virulent *Vp_AHPND_* strain 19-021-D1 and mutant strain 20-082A3.

pVA1-like Plasmid	Length (bp)	No. of CDSs	No. of SSRs *	AHPND Toxin	Tn3-like Transposon
pVp_Kor-D1-2	68,848	81	4	AHPND (*pirA, pirB*)	−
pVp_20-082A3B	67,093	75	4	AHPND mutant (*pirB*, partial)	−

* SSRs, Short Sequence Repeats (5′-TTGTTTTTC-3′).

## Data Availability

All data generated or analyzed in this study are available in this manuscript and its Appendix A.

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
