# Peer review of "Genomic and Pathological Characterization of Acute Hepatopancreatic Necrosis Disease (AHPND)-Associated Natural Mutant Vibrio parahaemolyticus Isolated from Penaeus vannamei Cultured in Korea"

_animals, 2024, doi:10.3390/ani14192788_

Round 1
Reviewer 1 Report
Comments and Suggestions for Authors
General comment
Well planned and executed study adding significant value to scientific literature on this pathogen of high economic importance to shrimp.
The authors have identified and isolated a natural type II (pirA-, pirB+) mutant Vibrio parahaemolyticus strain from P. vannamei shrimp cultures in Korea. After molecular characterisation of the toxin status, hemolytic activity, antimicrobial profile, and taxonomic affiliation of the Vibrio strain using published PCR assays and agar plate culture, the authors used PacBio long-read and Illumina short-read approach to sequence the whole genome of the mutant Vibrio strain for in-depth in-silico molecular typing and genome analysis and annotation.
Overall, this is a comprehensive and rounded study combining molecular genomics and functional phenotypic analysis. The presented study will add important insight to the literature about Vibrio, AHPND, and overall understanding of plasmid borne toxin characteristics for this important shrimp pathogen.
It is well written and requires minor proof-reading attention.
All molecular data (PCR- and whole genome based) are very well presented, agar plate culture related results and antimicrobial data are reasonably well presented, but in absence of any protein-biochemistry functional results (transcript detection and/or protein expression or other functional assays), the presentation of the shrimp bioassay data about the pathogenic activity of the mutant versus the virulent strain are not very well presented.
I suggest that the authors expand on the bioassay data and list observations (e.g. animals survival animal mortality comparison) and ideally a brief validation assay / test about presence and absence of the suspected plasmid or strain in the challenged shrimp cultures to proof the challenge dose was sufficient for mutant strain to be detectable.
Specific comments
Line 21 causing should not be plural and should be causes or you could say is causing
Line 23 missing is
Line 27 insert comma Korea, and also
Line 22/23/24 this sentence is poorly worded and needs changing
“From the first AHPND outbreak in culturing Penaeus vannamei in Korea at 2016, the AHPND-associated Vibrio parahaemolyticus (VpAHPND) still considered major threat to the Korean shrimp industry.”
Suggestion: “Since the first AHPND outbreak in P. vannamei cultures in Korea in 2016, AHPND-associated Vibrio parahaemolyticus (VpAHPND) is still considered a major threat to the Korean shrimp industry.”
Line 25: “have been raised on the misdiagnosis” change to “…raised about the …”
Line 391 – 396
Please expand and consider adding testing results to confirm mutant strain challenge dose was sufficient and strain or plasmid still detectable in culture after xxx days.
Questions
Line 205
Were the shrimp collected from the farm tested to ensure they were negative for the pathogen used in the bioassay before starting.
Recommendation
Accept after minor review
Comments on the Quality of English Languageabove
Author Response
-Reviewer 1:
We would like to thank reviewer 1 for the helpful comments and suggestions. We have revised the manuscript based on the comments and have provided our point-by-point responses as follows:
[General Comment]
Well planned and executed study adding significant value to scientific literature on this pathogen of high economic importance to shrimp. The authors have identified and isolated a natural type II (pirA-, pirB+) mutant Vibrio parahaemolyticus strain from P. vannamei shrimp cultures in Korea. After molecular characterisation of the toxin status, hemolytic activity, antimicrobial profile, and taxonomic affiliation of the Vibrio strain using published PCR assays and agar plate culture, the authors used PacBio long-read and Illumina short-read approach to sequence the whole genome of the mutant Vibrio strain for in-depth in-silico molecular typing and genome analysis and annotation. Overall, this is a comprehensive and rounded study combining molecular genomics and functional phenotypic analysis. The presented study will add important insight to the literature about Vibrio, AHPND, and overall understanding of plasmid borne toxin characteristics for this important shrimp pathogen. It is well written and requires minor proof-reading attention.
All molecular data (PCR- and whole genome based) are very well presented, agar plate culture related results and antimicrobial data are reasonably well presented, but in absence of any protein-biochemistry functional results (transcript detection and/or protein expression or other functional assays), the presentation of the shrimp bioassay data about the pathogenic activity of the mutant versus the virulent strain are not very well presented. I suggest that the authors expand on the bioassay data and list observations (e.g. animals survival animal mortality comparison) and ideally a brief validation assay / test about presence and absence of the suspected plasmid or strain in the challenged shrimp cultures to proof the challenge dose was sufficient for mutant strain to be detectable.
Response> Thank you for the valuable comment. As commented by the reviewer, we added more information for bioassay data (mortality comparison, PCR result) in the text, as the following.
1. PCR result: “Experimental shrimp were fed with commercial shrimp feed at 5% of shrimp biomass, and survival was monitored and recorded for 7 days. For AHPND confirmation, representative (live or moribund) shrimp samples were fixed in 95% ethanol for PCR assays, and fixed in Davidson’s AFA (alcohol, formaldehyde, acetic acid) fixative for histopathology analysis, at 24 h post-infection from each group.” (Line 228-233, in Section 2.7.)
“PCR analysis of the shrimp confirmed the presence of the toxin genes (pirA and pirB) in positive control shrimp (hepatopancreas), whereas only pirB+ gene was detected in the shrimp (hepatopancreas) challenged by the mutant VpAHPND isolate, and the toxin genes were not detected in negative control shrimp at 24 h.” (Line 415-418, in results)
2. Mortality comparison: “Same with the shrimp in negative control (without bacteria), the survival rate of shrimp challenged by the mutant VpAHPND isolate was 100%, and the live shrimp showed normal external appearances and feeding behaviors on the termination day of the experiment (day 7). In contrast, the shrimp in positive control (challenged by the virulent VpAHPND strain, 13–028/A3) showed typical AHPND symptoms and became dead at 48-72 h.” (Line 409-414, in Section 3.5.)
[Specific Comments]
- Line 21 causing should not be plural and should be causes or you could say is causing
Response> Thank you for your valuable comment. According to the reviewer’s comment, we have revised the word from “causing” to “causes” in Line 21.
- Line 23 missing is
Response> Thank you for your valuable comment. According to the reviewer’s comment, we have added the word “is” in Line 23.
- Line 27 insert comma Korea, and also
Response> Thank you for your valuable comment. According to the reviewer’s comment, we have added the comma “Korea, and also” in Line 27.
- Line 22/23/24 this sentence is poorly worded and needs changing
“From the first AHPND outbreak in culturing Penaeus vannamei in Korea at 2016, the AHPND-associated Vibrio parahaemolyticus (VpAHPND) still considered major threat to the Korean shrimp industry.”
Suggestion: “Since the first AHPND outbreak in P. vannamei cultures in Korea in 2016,
AHPND-associated Vibrio parahaemolyticus (VpAHPND) is still considered a major threat to the Korean shrimp industry.”
Response> Thank you for your insightful suggestion. According to the reviewer’s comment, we have revised the sentence in Lines 22-23.
- Line 25: “have been raised on the misdiagnosis” change to “…raised about the …”
Response> Thank you for your insightful suggestion. According to the reviewer’s comment, we have revised the sentence in Line 25.
- Line 391 – 396: Please expand and consider adding testing results to confirm mutant strain challenge dose was sufficient and strain or plasmid still detectable in culture after xxx days.
Response> Thank you for your insightful suggestion. To be sure the mutant strain challenge dose was sufficient, we followed the bioassay protocol for AHPND mutant described by [12], and tested shrimp samples for AHPND duplex PCR assay at 24 h post-infection, to check whether those shrimps were infected by either mutant or VpAHPND strains. Additionally, we added PCR assay (Figure 4) and histopathology data (Figure 5) to confirm bacterial infection and revised the text as follows.
“Shrimps were immersed in water containing bacteria at a concentration of approximately 106 CFU/mL, “following the protocol for the detection of mutant VpAHPND [12]” (Line 224, in Section 2.7.)
“PCR analysis of the shrimp confirmed the presence of the toxin genes (pirA and pirB) in the shrimp HP of the positive control, whereas only the pirB gene was detected in the shrimp HP challenged by the mutant VpAHPND isolate (20-082A3), and the toxin genes were not detected in the shrimp HP of the negative control at 24 h (Figure 4). In addition, the natural mutant VpAHPND strain 20-082A3 did not cause any cumulative mortalities to the indicator P. vannamei and histopathological examination of the surviving shrimp showed bacterial colonization of the HP with light atrophy, which differed from those observed in the positive control group” (Figure 5). (Line 415-422, in Section 3.5.)
[Questions]
- Line 205: Were the shrimp collected from the farm tested to ensure they were negative for the pathogen used in the bioassay before starting.
Response> Thank you for your valuable comment. Yes, we have tested the shrimps collected from the farm to be sure they were negative for VpAHPND infection before bioassay. As commented by the reviewer, we added the information as follows. We also added a photo of the PCR assay in Figure 4.
“Before the bioassay, three shrimp were randomly selected and confirmed that the tested shrimp (hepatopancreas, HP) did not carry pirA and/or pirB genes using a duplex PCR assay as described previously [2].” (Line 213-215, in Section 2.7.)
We hope these revisions are satisfactory and we thank you very much for helping us to improve this manuscript.
Sincerely,
Ji Hyung Kim

Reviewer 2 Report
Comments and Suggestions for Authors
The manuscript highlights the occurrence of natural deletion mutants of AHPND causing V. parahaemolyticus similar to what has been already reported from other places. However, the authors need to address the following queries and suggestions, which would enhance the report presented in this manuscript.
1. The sequenced strain was obtained from a suspected V. parahaemolyticus infection in a shrimp farm. Did the sampled shrimp exhibit typical AHPND signs?
2. Was the V. parahaemolyticus outbreak due to natural mutant , or were there isolates lacking the pVA-like plasmids as well? Since only one mutant isolate was chosen for sequencing, it remains to be clarified if there were multiple strains involved in the outbreak?
3. Though shrimp bioassay was performed, it is unclear as to why the results of the same were not included in the manuscript. Did histopathological observations from shrimp bioassay exhibit similar observations as that of the hepatopancreas from which the mutant was isolated?
4. Since the histopathological observation differed for the control and test groups, the results need to be presented and discussed.
5. Varying salt concentration ( 2 % and 3% ) has been used for different experiments. Why did the authors chose to use different NaCl (%) supplement for bacterial culture growth/maintenance and AST experiment? The preferred concentration is 1% for growth/ maintenance and AST.
6. Was both T6SS1 and T6SS2 absent in the sequenced genome? Since VpT6SS2 is present in all V. parahaemolyticus isolates, it is unusual to see no mention of T6SS2 in the sequenced genome (VFDB and annotation results).
The authors are advised to look for this in their sequence.
7. Have authors checked for classical pathogenicity islands VPaI 1-7 in the sequenced genome?
8. Did the enlisted prophage sequences encode any potential virulence genes that could have enhanced the pathogenicity of the sequenced strain?
Discussion on the above points would help improve the manuscript.
Author Response
-Reviewer 2:
We would like to thank reviewer 2 for the helpful comments and suggestions. We have revised the manuscript based on the comments and have provided our point-by-point responses as follows:
[Comments and Suggestions for Authors]
The manuscript highlights the occurrence of natural deletion mutants of AHPND causing V. parahaemolyticus similar to what has been already reported from other places. However, the authors need to address the following queries and suggestions, which would enhance the report presented in this manuscript.
- The sequenced strain was obtained from a suspected V. parahaemolyticus infection in a shrimp farm. Did the sampled shrimp exhibit typical AHPND signs?
Response> Thank you for your valuable comments. The mutant AHPND Vibrio strain (20-082A3) was obtained from the moribund shrimp showing typical symptoms of AHPND including soft shell, whitish hepatopancreas, and empty gut. However, the shrimp challenged by the strain 20-082A3 did not show any clinical signs in bioassay. In the collected samples, typical VpAHPND strains were also isolated. We have revised the sentences in Section 3.1 to make them clearer.
- Was the V. parahaemolyticus outbreak due to natural mutant , or were there isolates lacking the pVA-like plasmids as well? Since only one mutant isolate was chosen for sequencing, it remains to be clarified if there were multiple strains involved in the outbreak?
Response> Thank you for your valuable comments. As mentioned previously, the mutant AHPND strain (20-082A3) was obtained together with typical VpAHPND strains from the moribund shrimp showing AHPND symptoms including soft shell, whitish hepatopancreas, and empty gut, but the shrimp challenged by the strain 20-082A3 did not show any clinical signs in bioassay. During the surveillance study, several Vibrio strains were isolated from the farm samples, as well as the mutant AHPND strain (20-082A3). However, only the mutant VpAHPND strain was chosen for further study (sequencing and shrimp bioassay). (Line 244-248, In Section 3.1.)
- Though shrimp bioassay was performed, it is unclear as to why the results of the same were not included in the manuscript. Did histopathological observations from shrimp bioassay exhibit similar observations as that of the hepatopancreas from which the mutant was isolated?
Response> Thank you for your valuable comments. As commented by the reviewer, we added the results of the bioassay study (section 3.5. Shrimp bioassays) (Line 406-422).
- Since the histopathological observation differed for the control and test groups, the results need to be presented and discussed.
Response> Thank you for your valuable comments. As commented by the reviewer, we newly added histopathology results in the text (Line 418-422, In Section 3.5.) and Figure 5.
- Varying salt concentration (2 % and 3%) has been used for different experiments. Why did the authors chose to use different NaCl (%) supplement for bacterial culture growth/maintenance and AST experiment? The preferred concentration is 1% for growth/ maintenance and AST.
Response> Thank you for your valuable comments. As stated in the manuscript, we followed the methodology described in the cited reference 27 (Siriphap et al., 2024) for the antimicrobial susceptibility test (AST) experiment, which used a 3% NaCl concentration. We acknowledge that the preferred NaCl concentration for bacterial growth, maintenance, and AST experiments is generally 1%. However, we performed the protocol as previously described to maintain the consistency of our findings with the referenced study.
[REF 27] Siriphap, A.; Prapasawat, W.; Borthong, J.; Tanomsridachchai, W.; Muangnapoh, C.; Suthienkul, O.; Chonsin, K. Prevalence, virulence characteristics, and antimicrobial resistance of Vibrio parahaemolyticus isolates from raw seafood in a province in Northern Thailand. FEMS Microbiol. Lett. 2024, 371.
- Was both T6SS1 and T6SS2 absent in the sequenced genome? Since VpT6SS2 is present in all V. parahaemolyticus isolates, it is unusual to see no mention of T6SS2 in the sequenced genome (VFDB and annotation results). The authors are advised to look for this in their sequence.
Response> Thank you for your valuable suggestion. Although T6SS1 and T6SS2 were not detected in the initial VFDB analysis results, we have conducted additional in silico genomic analysis of V. parahaemolytics strains 20-082A3 and 19-021-D1 to identify the presence of these secretion systems. Our analysis was based on a previously reported study (Jana et al., 2022), where T6SS1 and T6SS2 were identified in the complete genome sequence of V. parahaemolyticus RIMD 2210633. By comparing two strains to this reference, we have confirmed the presence of T6SS1 in the chromosome 1 sequences of both strains 20-082A3 and 19-021-D1, while T6SS2 was found in their respective chromosome 2 sequences. We have added a description of the T6SS1 and T6SS2 identification in Lines 320-325 and Lines 502-503 of the revised manuscript and Table S2.
[REF] Jana, B., Keppel, K., Fridman, C. M., Bosis, E., Salomon, D. Multiple T6SSs, mobile auxiliary modules, and effectors revealed in a systematic analysis of the Vibrio parahaemolyticus pan-genome. Msystems, 2022. 7(6), e00723-22.
- Have authors checked for classical pathogenicity islands VPaI 1-7 in the sequenced genome?
Response> Thank you for your insightful suggestion. According to the reviewer’s comment, we have conducted a genomic analysis of V. parahaemolytics strains 20-082A3 and 19-021-D1 genomes to identify the presence of genomic islands (GIs), including putative pathogenicity islands VPaI 1-7. Our analysis was based on previously reported study (Hurley et al., 2006), where GIs were identified in the complete genome sequence of V. parahaemolyticus RIMD 2210633. We have added a description of the VPaI identification in Lines 325-328 and Lines 502-503 of the revised manuscript and Table S3.
[REF] Hurley, C. C., Quirke, A., Reen, F. J., Boyd, E. F. Four genomic islands that mark post-1995 pandemic Vibrio parahaemolyticus isolates. BMC genomics, 2006. 7.
- Did the enlisted prophage sequences encode any potential virulence genes that could have enhanced the pathogenicity of the sequenced strain?
Response> Thank you for your helpful comment. According to the reviewer’s comment, we confirmed potential virulence genes in putative four prophage regions (region Ⅰ-Ⅳ listed in table S3) in strain 20-082A3 genome, using VFDB (http://www.mgc.ac.cn/VFs/) and BLASTp search to identify any potential virulence genes that might contribute to the pathogenicity of this strain. Specifically, we screened the four representative prophage regions against virulence genes from the Vibrio genus classified in the VFDB database. Additionally, we performed a BLASTp search against the sequences of the classical seven genomic islands (VPaI 1-7), and the Type 6 secretion system (T6SS1 and T6SS2) gene clusters from the reference strain V. parahaemolyticus RIMD 2210633 (chromosome 1, NC_004603.1; chromosome 2, NC_004605.1). The VFDB and BLASTp analysis did not reveal any significant matches to known virulence genes within the prophage regions of strain 20-082A3.
We hope these revisions are satisfactory and we thank you very much for helping us to improve this manuscript.
Sincerely,
Ji Hyung Kim
